# Surface Treatment of Acetabular Cups with a Direct Deposition of a Composite Nanostructured Layer Using a High Electrostatic Field

**DOI:** 10.3390/molecules25051173

**Published:** 2020-03-05

**Authors:** Marek Pokorný, Tomáš Suchý, Adéla Kotzianová, Jan Klemeš, František Denk, Monika Šupová, Zbyněk Sucharda, Radek Sedláček, Lukáš Horný, Vlastimil Králík, Vladimír Velebný, Zdeněk Čejka

**Affiliations:** 1Contipro a.s., R&D Department, 561 02 Dolni Dobrouc, Czech Republic; kotzianova@contipro.com (A.K.); klemes@contipro.com (J.K.); velebny@contipro.com (V.V.); 2Department of Composites and Carbon Materials, Institute of Rock Structure and Mechanics, Czech Academy of Sciences, 180 00 Prague 8, Czech Republic; suchy@irsm.cas.cz (T.S.); denk@irsm.cas.cz (F.D.); supova@irsm.cas.cz (M.Š.); sucharda@irsm.cas.cz (Z.S.); 3Faculty of Mechanical Engineering, Czech Technical University in Prague, 166 00 Prague 6, Czech Republic; radek.sedlacek@fs.cvut.cz (R.S.); lukas.horny@fs.cvut.cz (L.H.); or vlasta.kralik@lfp.cuni.cz (V.K.); 4ProSpon s.r.o., 272 01 Kladno, Czech Republic; zdenek.cejka@prospon.cz

**Keywords:** collagen composite, hydroxyapatite, titanium implant, nanofibers, electrospinning

## Abstract

A composite nanofibrous layer containing collagen and hydroxyapatite was deposited on selected surface areas of titanium acetabular cups. The layer was deposited on the irregular surface of these 3D objects using a specially developed electrospinning system designed to ensure the stability of the spinning process and to produce a layer approximately 100 micrometers thick with an adequate thickness uniformity. It was verified that the layer had the intended nanostructured morphology throughout its entire thickness and that the prepared layer sufficiently adhered to the smooth surface of the model titanium implants even after all the post-deposition sterilization and stabilization treatments were performed. The resulting layers had an average thickness of (110 ± 30) micrometers and an average fiber diameter of (170 ± 49) nanometers. They were produced using a relatively simple and cost-effective technology and yet they were verifiably biocompatible and structurally stable. Collagen- and hydroxyapatite-based composite nanostructured surface modifications represent promising surface treatment options for metal implants.

## 1. Introduction

Research conducted in the past century revealed that the most suitable artificial hard tissue implants (especially bones and large joints) would require a combination of a metal body and a surface treatment with a hydroxyapatite (HAp) and collagen (COL) composite. It was also proven that in order to improve osseointegration, the surface layer must suitably imitate the porous micro- or nanostructure of the respective tissue (bone or the surrounding tissue) [1]. At the time, no technology had been developed that would allow for such a delicately structured composite layer to be produced directly on the surface of a metal implant. Today, the most frequently studied types of methods are physical and chemical [2,3], and biological methods are receiving increasing attention as well [4]. Collagen as well as collagen and hydroxyapatite composites are currently also processed into nanostructures [5,6] using the electrostatic spinning method [7]. Nanofibrous structures containing HAp are prepared from solutions of polymers, both natural, in this case collagen, and synthetic. Synthetic polymers in such solutions mostly serve an auxiliary role by promoting fiber formation, but they can also influence the characteristics of the resulting structure. The most frequently used synthetic polymers are polycaprolactone [8], polylactid acid [9], and polyvinyl alcohol [10].

However, the electrostatic spinning (ES) method was designed primarily for the deposition of nanofibrous layers onto flat 2D collectors. In its simplest form, an ES system consists of two non-moving electrodes—a spinning jet and a collector. In order to prepare larger-area structures with a homogeneous distribution of nanofibers, the electrodes must be moving. At present, this requirement is fulfilled by moving the spinning jet along a line and by rotating the collector [11]. Nevertheless, achieving thickness homogeneity on practically applicable 2D surfaces remains problematic [12]. One of the major advantages of ES is its great variability, which allows the spinning system to be modified in such a way as to allow depositing nanofibers on a three-dimensional object of a simple shape, as described, for example, in this paper [13]. Even though the methods for the characterization and testing of surface treatments for implants described in available scientific literature refer only to flat 2D shapes simulating the surface of actual implants [14], the presented results are very promising as to their future use in orthopedic practice [15]. Unfortunately, results for real 3D implants with different titanium surfaces are still unavailable. What is more, there is no available scientific literature that describes any deposition method of direct application of nanofibers onto the surface of an implant. Such a method is described only in a patent application [16] concerning surface treatments of dental implants. The simplified procedure described in the patent application [17] involves coating the surface of an implant with a pre-prepared nanofibrous layer.

This paper deals with the preparation of composite nanofibrous layers composed of COL/HAp/PEO (polyethylene oxide) using the electrostatic spinning method. The composition and structure of the composite layer and its suitability for its intended application, i.e., implant surface modification, as well as the suitability of its kinetic drug release properties, were tested and verified in our previous research [18,19]. Moreover, the cytocompatibility of electrospun nanofibrous COL/HAp layers was evaluated in our previous studies using human osteoblastic cells in direct contact with the layers or in 24-h infusions thereof [18,19]. No cytotoxic effects were indicated in the case of layers without antibiotics as well as in the case of layers impregnated with vancomycin or combination of vancomycin and gentamicin. A dispersion containing the abovementioned substances was applied directly onto the surface of model implants in the form of acetabular cups, which differed from real implants only in their smooth surface (not roughened). A new collector was designed for this purpose to allow several implants to be coated simultaneously. Furthermore, the dispersion inside the dosing syringe was continuously mixed to ensure the process would be stable over the entire duration of deposition and to prevent changes in nanofiber morphology due to the sedimentation and separation of the individual components of the dispersion, see Figure 1. The aim was to use the electrostatic spinning method to produce a surface layer with a uniform thickness of 100 micrometers in a selected area. To achieve this result, horizontally-moving spinning jets were placed under vertically-rotating implants attached to a collector electrode. After its deposition, the material was analyzed to verify its porous nanofibrous structure and the desired chemical composition, and that the produced layer would retain sufficient adhesion to the implant surface after its subsequent processing, including treatments resulting in the cross-linking of collagen and drying of the deposited layer.

## 2. Materials

The prepared spinning solution was a dispersion of hydroxyapatite nanoparticles (average size of 150 nm, Sigma Aldrich) in a blend of collagen type I (VUP Medical a.s., Brno, Czech Republic) and polyethylene oxide (900 kDa, Sigma Aldrich, St. Louis, MO, USA) in a mixture of ethanol (Penta, Prague, Czech Republic) and a phosphate buffer solution (PBS, Sigma Aldrich, St. Louis, MO, USA) with a volume ratio of 1:1. The total concentration of each component in spinning solution was 1.38 wt.% HAp, 7.84 wt.% COL, 0.63 wt.% PEO, and 90.15 wt.% ethanol/PBS. The solution was mixed with a homogenizer before the spinning process. A solution of ethanol/H_2_O/EDC/NHS, where EDC stands for *N*-(3-Dimethylaminopropyl)-*N*′-ethylcarbodiimide hydrochloride (Sigma Aldrich, St. Louis, MO, USA) and NHS stands for *N*-hydroxysuccinimide (Sigma Aldrich, St. Louis, MO, USA), was used for the cross-linking of the nanofibrous layer after electrospinning. Finally, layers were extensively washed in 0.1M Na_2_HPO_4_ and H_2_O, frozen at −15 °C and lyophilized (BenchTop 4KZL, VirTis, Warminster, PA, USA). In this step, PEO was fully leached out, so the final concentration of COL/HAp in the layer was 85/15 (*w*/*w*). The preparation of spinning solution is described in detail elsewhere [18].

## 3. Methods

### 3.1. Electrospinning

A 4SPIN LAB device (Contipro a.s., Dolni Dobrouc, Czech Republic) was used for the spinning process. A 30-mL syringe with the prepared dispersion was placed into the device’s dosing system, from which the mixture was subsequently dosed at a rate of 70 µL/min. During the entire process, the syringe was rotated 180° around its axis every 30 s. The solution was dosed through two needles (G19, Hamilton, Reno, NV, USA) fastened 12 cm apart. Both needles moved horizontally from side to side along a linear 20 cm path at a speed of 10 cm/s. Both needles were connected to a positive voltage of 40 kV. Four model acetabular cup implants were attached 20 cm above the needles and continuously rotated vertically at a rate of 10 rpm. During the entire 6-h-long process, the ambient temperature was kept at 25 ± 3 °C and humidity at (22 ± 5) %RH.

### 3.2. Numerical Simulation

The process parameters, i.e., the geometry of the experiment, were set based on numerical simulations of the distribution of electrostatic field intensities performed in the Comsol software (Burlington, MA, USA). The results of these electrostatic field distribution simulations are shown in Figure 2.

### 3.3. Morphology Analysis

The morphology of the nanofibrous layers was studied using a Zeiss ULTRA PLUS scanning electron microscope (ZEISS, Oberkochen, Germany). The samples were coated with a very thin layer of Au and Pd in a Leica EM ACE600 coater (Leica, Wetzlar, Germany). Images were acquired using an SE detector; the working distance ranged from 4 mm to 6 mm, and the accelerating voltage was 3.5 kV. Randomly selected images representative of the entire implant area covered in a nanofibrous layer were used to measure the diameters of 80 individual nanofibers in the ImageJ software and the results were represented with a histogram and defined with mean +/- SD. The deposited nanofibrous layers were evaluated first immediately after the spinning process and again after their cross-linking and the subsequent lyophilization in order to determine how these processes affect the morphology of deposited nanofibers. Furthermore, the ability of the deposited material to retain its fibrous structure was assessed.

### 3.4. Qualitative Analysis

Raman spectroscopy was used to perform a qualitative analysis. The measurements were performed on an inVia^TM^ Qontor device (Renishaw, Wotton-under-Edge, United Kingdom) equipped with a laser with a wavelength of 532 nm. The samples were measured untreated. The exposure time was set to 5 s with 10 accumulations. An objective with a magnification of 20 was used. All spectra were analyzed in the WiRE software (Renishaw, Wotton-under-Edge, United Kingdom) and Origin software (OriginLab, Northampton, MA, USA).

### 3.5. Determination of Layer Thickness

The geometry of each implant was precisely measured using a contactless optical coordinate measuring machine (Redlux Ltd., Romsey, United Kingdom). Selected surface areas were measured at three levels (at heights of 3, 13, 23 mm) corresponding to three different diameters of the implant (Figure 1B); at least 1000 points/rotation were measured each time. Implant geometry was measured repeatedly in the following three stages of layer preparation: (1) before the deposition of the layer, (2) after the deposition of the layer, and (3) after the cross-linking and drying of the deposited layer to determine the effects of cross-linking and lyophilization. The obtained data were used to determine layer thickness and layer thickness uniformity across the surface of each implant.

### 3.6. Determination of Adhesion Strength

The adhesion between the electrospun COL/HAp layer and titanium surface was determined by means of adaptation of the shaft-loaded blister test [20] (see Figure 3). The adhesion was quantified based on calculating the maximum bond stress required for layer separation. Prior to testing, the surfaces of samples were chemically treated in order to evaluate the effect of four different methods for the adhesion improvement. The first group of samples (Ac) was degreased by acetone in ultrasound bath (UB) for 10 min. The second group was (Ac-PBS) degreased (acetone, 10 min, UB), immediately followed by immersion in PBS (10min), rinsed with deionized H_2_O, and dried in a hood. Samples from the third group (Ac-HF-COL) were degreased (acetone, 10min, UB), etched for 2 min in solution of 5 g Na_3_PO_4_, 0.9 g NaF, 1.6 g (50 wt% HF) supplemented by water up to 100 g, rinsed with H_2_O (5 min) followed by fast drying (70 °C) and immediately impregnated by diluted collagen/water solution (1/10, *w*/*w*) containing EDC/NHS (4/1, *w*/*w*). After cross-linking in situ, samples were washed in the 0.1 M Na_2_HPO_4_ (2 × 15 min), followed by rinsing using deionised water (5 min) and dried in a hood. Finally, samples from the fourth group (Ac-COL) were degreased (acetone, 10 min, UB), dried impregnated by diluted collagen/water solution (1/10, *w*/*w*) containing EDC/NHS (4/1, *w*/*w*), and washed after cross-linking as described above (Ac-HF-COL). Untreated samples served as controls (0). After these procedures, all samples (each group *n* = 10) were immediately coated with collagen by means of electrospinning (1 h) of collagen solution, as described above.

### 3.7. Statistical Evaluation

The statistical analysis was performed using statistical software (STATGRAPHICS Centurion XVII, StatPoint, The Plains, WV, USA). For the data presented in the form of box-and-whiskers plots, a box was drawn extending from the lower quartile to the upper quartile of the sample (this interval covers the middle 50% of the values sorted from smallest to largest). A vertical line was drawn at the median and a plus sign was drawn at the sample mean (arithmetical). Whiskers were drawn from the edges of the box to the largest and smallest data values unless values were situated unusually far from the box. Point symbols outside the whiskers indicate values which were >1.5 times the interquartile range (box width) above or below the box. All points >3 times the interquartile range above or below the box were termed far outside points and are indicated by point symbols with plus signs superimposed above them. In the case of the presence of outside points, the whiskers were drawn to the largest and smallest data values which do not constitute outside points. The normality of the data was verified primarily by means of the Shapiro–Wilk’s test. Homoscedasticity was verified by means of the Levene’s test. The Fisher’s least significant difference (LSD) procedure was applied for a multiple sample comparison. Statistical significance was accepted at *p* ≤ 0.05.

## 4. Results and Discussion

To achieve a homogeneous coating of a 3D object in the 4SPIN LAB device, we developed a new collector, to which the acetabular cups can be attached (four implants in a line). The new collector can be simply installed/inserted in 4SPIN LAB—no modification of the device itself is required. Moreover, the design of the new collector allows the attached acetabular cups to be vertically rotated during the electrospinning process. We also used a system that allows the spinning electrodes to move linearly along a horizontal path under the collector in order to deposit uniform nanofibrous layers on all the implants. This system can also be easily installed in the 4SPIN LAB device and requires no modifications. The optimum distance between the electrodes and the optimum distance between the two spinning jets, i.e., needles, as well as the optimum line of movement of the spinning jets were set based on the electrostatic field distribution simulations carried out in the Comsol environment. The rate of rotation of the collector and the speed of linear movement of the spinning jets were set based on the results of an optimization process performed in advance. We had to ensure the stability of the long spinning process and an even distribution of HAp in the spinning dispersion as well as in the deposited nanofibrous layers. In other words, an issue with the sedimentation of the solid component of the spinning dispersion needed to be resolved. It was observed that the sedimentation of the solid component of the dispersion began after a mere 30 min, making it clear that the solution would need to be regularly mixed during the entire electrospinning process without interrupting it. This issue was resolved by using a mechanism, which periodically rotated the syringe in the 4SPIN LAB dosing system 180 degrees around its axis every 30 s. This interval and degree of rotation proved optimal as even after several hours of spinning, there was no visible separation of the solid component from the dispersion and the morphology of nanofibers was identical after 2, 4, and 6 h (these results are not presented here).

In the studied coated areas, each of the deposited layers was measured to be on average more than 113-µm thick, except for sample number 2, which was coated with a layer less than 70-µm thick despite the same time of deposition, see Table 1. In all cases, deviations in layer thickness along the circumference of each implant measured at the three scanning levels were less than 10%. This implies that the thickness uniformity of the deposited nanofibrous layers was high all over the coated implants, see Figure 4. On the other hand, there were differences between the average values of layer thickness measured at the individual scanning levels; for example, in the case of sample 4, the difference between values measured from 3 mm and 23 mm was as high as 80 µm. This was reflected in the value of the standard deviation of the calculated overall layer thickness of this sample, which was ±31.6 µm (±28%), the highest of all the studied samples. In the cases of samples 3 and 4, the layers covering smaller-diameter sections of the coated implants had a lower thickness, which was a logical outcome as due to the smaller diameters, the surfaces of these sections were further away from the spinning jets, and fibers land preferentially on the surface area of the grounded collector closest to the jets. The results for the other two samples (number 1 and 2) were different, probably due to the electrostatic field being uneven, i.e., due to imperfections of the electrospinning system setup. Differences in the measured values of thickness were more pronounced after collagen cross-linking and lyophilization. Layer thicknesses increased in all cases by 2% to 55% compared to their initial values. The final average layer thickness was therefore higher than approximately 140 µm (again except for sample 2, where the post-deposition treatments had almost no effect on layer thickness) and the standard deviation ranged from 25% to 80%.

An analysis of electron microscope images taken after the deposition of the layers and after their cross-linking and lyophilization confirmed that fiber diameters increased after these treatments, see Figure 5. The results revealed that on average, fiber diameters increased by 15%, i.e., from 145 nm to 170 nm on average. The nanostructure was preserved throughout the entire thickness of each layer as can be seen on images of layer cross sections in Figure 5C,E.

After evaluating the obtained results, it can be concluded that the overall geometry of the deposited layers increased due to various environmental influences acting on the layers during the cross-linking of collagen and layer lyophilization. Besides the observed increase in nanofiber diameters, the overall layer thickness increased also due to an increased internal stress within the layer, which causes its localized separation from the implant surface, and because the pores between fibers/layers grew in size, compare Figure 5C,E. This also resulted in larger deviations in layer thickness detected in the samples. Nevertheless, the surface and thickness irregularities are acceptable for the intended application as the deposited composite material retained its composition (the content of COL/HAp) after subsequent processing and because in its intended application, the layer would also be subject to uneven loads and compressive stresses.

Raman spectroscopy was used to verify the content of the individual composite components in the deposited layers. Both COL and HAp were detected in the composite layers. The auxiliary polymer PEO was not detected as it was washed out during the cross-linking process. Nevertheless, the presence of PEO is not desirable for the intended application of the layer. Raman spectra of the composite structure (Figure 6) show numerous clearly visible bands associated with COL: Amide I (ν(C=O), 1664 cm^−1^), amino acids side chains (δ(CH_2_, CH_3_), 1450 cm^−1^), Amide III (ν(CN), δ(NH), 1275 cm^−1^), Phenylalanine (ν(CC), 1001 cm^−1^), α helix (ν(CC), 938 cm^−1^), and Proline (ν(CC), δ(CCH), 870 cm^−1^) [21]. The presence of HAp is confirmed by a single band in the 700–1900 cm^−1^ region; the band, associated with the ν(PO_4_) vibration group, is located at 960 cm^−1^ [22]. The intensity of this band is directly proportionate to the amount of HAp present, and as the solution contains HAp in the form of particles, the intensity of the band fluctuates due to the uneven distribution of HAp in the composite. However, this uneven distribution does not affect the suitability of the material for its intended application.

Various chemical treatments had different effects on adhesion improvement (see Figure 7). Very simple methods such as degreasing or degreasing/PBS immersion can significantly improve adhesion between electrospun collagen layers and plasma-sprayed substrates. The expected effect of etching was demonstrated, while impregnation did not have any effect. These results suggest that adhesion between electrospun collagen/antibiotic layers and plasma-sprayed titanium surfaces can be improved with chemical treatments of titanium surfaces.

## 5. Conclusions

A nanostructured composite layer containing collagen and hydroxyapatite was produced on the surface of model acetabular cups, which are used in artificial hip replacements. The layer was spun from a dispersion and deposited directly onto the titanium implant surface in a high electrostatic field. Even though the deposition method had to be modified and expanded by introducing moving spinning jets and a rotating collector carrying four implants, it remains a relatively simple, single-step, and efficient method of applying surface treatments to three-dimensional objects such as orthopedic implant components. Due to these modifications, it was possible to coat selected surface areas of all four model acetabular cups with layers of sufficient thickness and good thickness uniformity in a single process. Ensuring the stability of the spinning process made it possible to achieve the required thickness of more than 0.1 mm. The total degree of degradation of the original form of collagen after all steps of the process was insignificant. The performed tests of adhesion of the deposited layers to the smooth implant surface were not intended to evaluate their adhesion for the purposes of actual implant application, for which the implant surface is roughened, and the results, although evaluated as satisfactory, are therefore considered informative only of smooth surface adhesion. The methods and individual steps of surface treatments for artificial joint replacement components presented in this paper produce promising results applicable in the field of modern orthopedics.

## Figures and Tables

**Figure 1 molecules-25-01173-f001:**
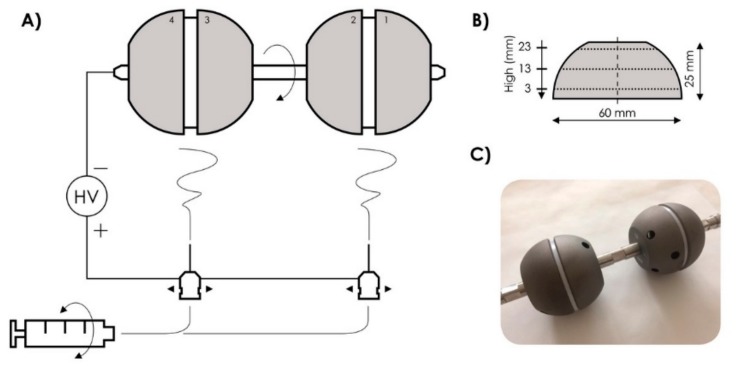
Diagram of the experimental deposition method (**A**). Dimensions of model acetabular cups and levels, at which the thickness of the deposited layer was measured (**B**). An image of four implants attached to the shaft of the rotary collector (rotary electrode).

**Figure 2 molecules-25-01173-f002:**
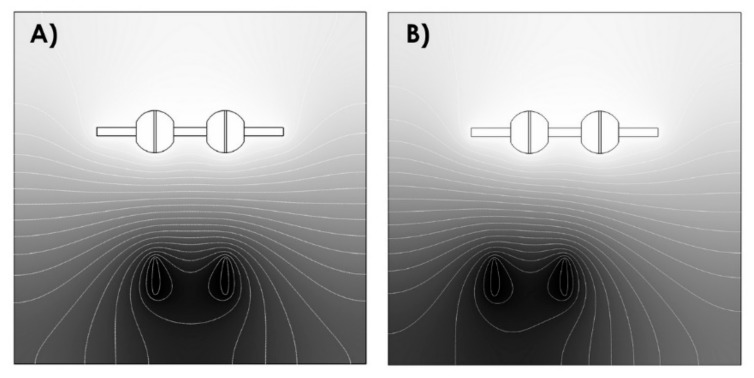
Results of numerical simulations of the distribution of electrostatic field intensities for a symmetrical arrangement (**A**) and for spinning jets at the left end of their path (**B**).

**Figure 3 molecules-25-01173-f003:**
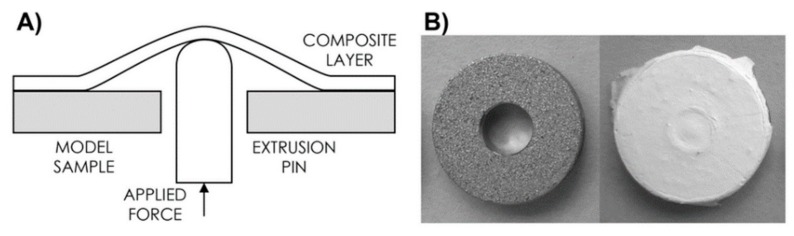
Diagram of the method used to measure adhesion of the composite layer to the model implant surface (**A**). Representative images of test samples before and after COL/HAp layer deposition (**B**).

**Figure 4 molecules-25-01173-f004:**
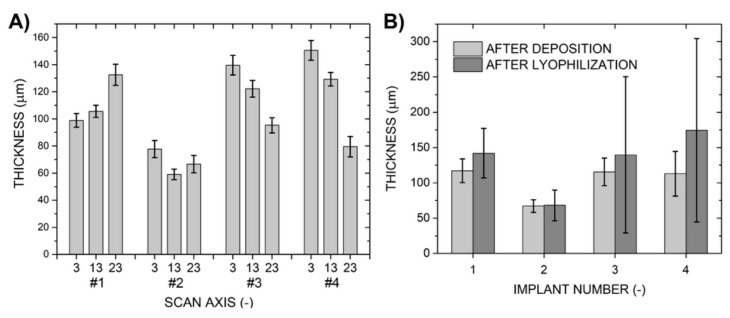
Layer thicknesses of the samples measured after their deposition at three different levels (**A**). Average layer thicknesses of individual samples after their deposition and subsequent lyophilization (**B**). The data used in these graphs are recorded in Table 1.

**Figure 5 molecules-25-01173-f005:**
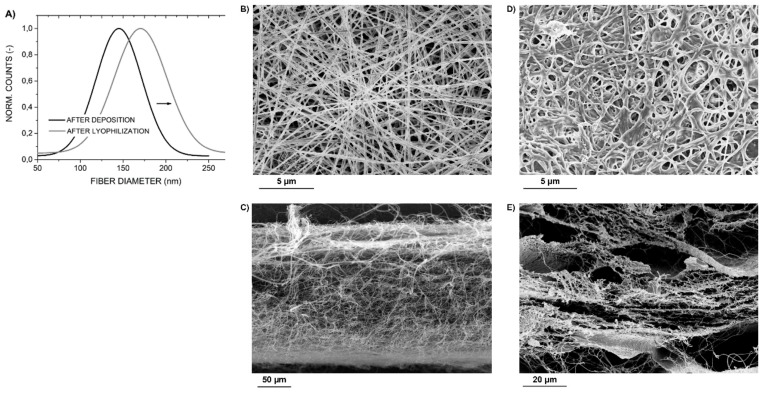
The distribution of fiber diameters after the deposition of the layer and after its cross-linking and lyophilization (**A**); corresponding SEM images of layer surfaces (**B**,**C**) and layer cross-sections (**D**,**E**).

**Figure 6 molecules-25-01173-f006:**
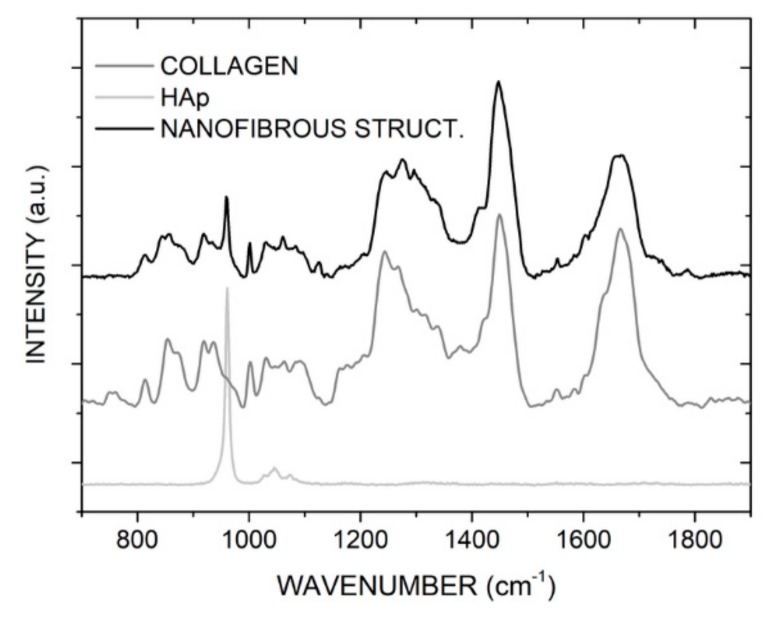
A Raman spectrum of the prepared composite material, including spectra of its individual components (COL, HAp).

**Figure 7 molecules-25-01173-f007:**
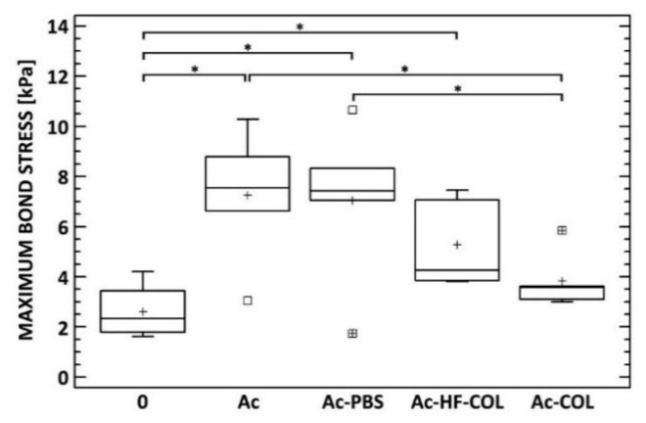
Box-plot of maximum bond stress required for separation of COL/HAp layer from differently treated titanium surfaces. * denotes statistically significant differences (Fisher’s LSD test, 0.05).

**Table 1 molecules-25-01173-t001:** Thicknesses of the deposited layers measured at three different levels along the height of the coated implants; average values for each sample after layer deposition and after lyophilization. The data is also shown in a graph included in Figure 4.

		After Deposition		After Deposition		After Lyophilization	
Implant No.	Scan axis (mm)	Layer thickness (µm)	Standard deviation (µm)	Layer thickness (µm)	Standard deviation (µm)	Layer thickness (µm)	Standard deviation (µm)
1	3	98.9	5.2	117.1	16.8	141.9	35.0
	13	105.6	4.4				
	23	132.5	7.8				
2	3	77.7	6.3	67.0	9.0	68.1	21.8
	13	59.1	3.9				
	23	66.7	6.4				
3	3	139.6	7.3	115.5	19.4	139.7	110.5
	13	122.2	6.1				
	23	95.2	5.6				
4	3	150.6	7.3	113.0	31.6	174.4	129.7
	13	129.2	5.0				
	23	79.5	7.5				

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
