# Peer review of "Surface Treatment of Acetabular Cups with a Direct Deposition of a Composite Nanostructured Layer Using a High Electrostatic Field"

_molecules, 2020, doi:10.3390/molecules25051173_

Round 1
Reviewer 1 Report
The authors coated the surface of titanium acetabular cups with a nanofibrous layer of nanocomposites consisting of collagen and hydroxyapatite nanoparticles via an electrostatic spinning method which is promising in hip replacement. This manuscript is well written and also well organized. Thus, the manuscript is publishable in this journal. The following suggestion should be addressed in the revised manuscript.
1. The biocompatibility of nanocomposite layers should be studied.
Author Response
Generally, the authors extend their thanks to reviewer for the valuable comments. When considering the revision of the original manuscript, we carefully considered the following points raised by the reviewer and modified the manuscript accordingly:
The biocompatibility of COL/HAp layers was evaluated in our previous studies (Suchý et al. 2017 and 2019; [18, 19]). Comprehensive in vitro tests were carried out using human osteoblastic cells (SAOS-2) in direct contact with the layers or in 24-hour infusions. Such layers were further tested after impregnation by antibiotics (vancomycin and gentamicin). We added this information to the last paragraph of Introduction.
Added text:
Moreover, the cytocompatibility of electrospun nanofibrous COL/HAp layers was evaluated in our previous studies using human osteoblastic cells in direct contact with the layers or in 24-hour infusions thereof [18, 19]. No cytotoxic effects was indicated in the case of layers without antibiotics as well as in the case of layers impregnated with vancomycin or combination of vancomycin and gentamicin.
Reviewer 2 Report
Dear Editor,
the article “Surface Treatment of Acetabular Cups with a Direct Deposition of a Composite Nanostructured Layer Using a High Electrostatic Field” by Marek et al. is interesting and is worth to be published, however after some minor revision.
Major requests:
Part b of Figure 4 is not in accordance with the data in table 1.
For example:
The layer thickness of Implant number 2 after deposition and after lyophilization is around 40 micron. In table 1 I can read 67 and 68 micron. Please set the axis of Figure 4 b in accordance.
The writing on the axes in Figures 4, 5 and 6 are too small. Please make them bigger.
Scale bars in SEM micrographs are not readable.
The SEM images are too small to show the preserved nanostructure throughout the entire thickness of each layer. Please make them bigger.
If contact angle measurements are available I suggest
Introduction needs revision as most of the references are 5 years old or more. Please consider recent reports in the field.
Based on the above arguments, I recommend the paper for publication after minor revision.
